# Influence of Metal Ions on Model Protoamphiphilic Vesicular Systems: Insights from Laboratory and Analogue Studies

**DOI:** 10.3390/life11121413

**Published:** 2021-12-16

**Authors:** Manesh Prakash Joshi, Luke Steller, Martin J. Van Kranendonk, Sudha Rajamani

**Affiliations:** 1Department of Biology, Indian Institute of Science Education and Research, Dr. Homi Bhabha Road, Pune 411008, Maharashtra, India; 2Australian Centre for Astrobiology, and School of Biological, Earth and Environmental Sciences, University of New South Wales, Kensington, NSW 2052, Australia; l.steller@unsw.edu.au (L.S.); m.vankranendonk@unsw.edu.au (M.J.V.K.)

**Keywords:** protocell, origins of life, prebiotic membranes, protoamphiphiles, metal ions, hot springs, N-acyl amino acid, analogue conditions

## Abstract

Metal ions strongly affect the self-assembly and stability of membranes composed of prebiotically relevant amphiphiles (protoamphiphiles). Therefore, evaluating the behavior of such amphiphiles in the presence of ions is a crucial step towards assessing their potential as model protocell compartments. We have recently reported vesicle formation by N-acyl amino acids (NAAs), an interesting class of protoamphiphiles containing an amino acid linked to a fatty acid via an amide linkage. Herein, we explore the effect of ions on the self-assembly and stability of model N-oleoyl glycine (NOG)-based membranes. Microscopic analysis showed that the blended membranes of NOG and Glycerol 1-monooleate (GMO) were more stable than pure NOG vesicles, both in the presence of monovalent and divalent cations, with the overall vesicle stability being 100-fold higher in the presence of a monovalent cation. Furthermore, both pure NOG and NOG + GMO mixed systems were able to self-assemble into vesicles in natural water samples containing multiple ions that were collected from active hot spring sites. Our study reveals that several aspects of the metal ion stability of NAA-based membranes are comparable to those of fatty acid-based systems, while also confirming the robustness of compositionally heterogeneous membranes towards high metal ion concentrations. Pertinently, the vesicle formation by NAA-based systems in terrestrial hot spring samples indicates the conduciveness of these low ionic strength freshwater systems for facilitating prebiotic membrane-assembly processes. This further highlights their potential to serve as a plausible niche for the emergence of cellular life on the early Earth.

## 1. Introduction

Membrane compartmentalization is a key feature of cellular life and is also thought to have been a crucial step towards the formation of first protocells on the prebiotic Earth. Contemporary cell membranes are primarily made of phospholipids, which are diacyl lipids. There have been few attempts to show prebiotically plausible chemical routes for phospholipid synthesis on the early Earth [1,2,3,4,5]. Nonetheless, phospholipids and other such structurally complex diacyl lipids are mostly considered as a product of membrane evolution that potentially occurred over protracted timescales of life’s evolution [6,7,8]. On the contrary, primitive membranes are thought to have been mainly composed of chemically simpler single chain amphiphiles (SCAs), such as fatty acids and their derivatives [7,9]. Given their plausible availability on the prebiotic Earth [10,11] and ability to self-assemble into membranes, these SCAs which presumably served as membrane components of early protocells can be collectively called as protoamphiphiles [12]. In general, the membrane-assembly of such protoamphiphiles is affected by several parameters like temperature, pH, and ionic strength of a medium [13,14,15,16]. Particularly, fatty acid vesicles are extremely sensitive to metal ions (especially divalent ions like Mg^2+^), which interact with their negatively charged carboxylate head groups and induce the formation of aggregates. However, metal ions have been shown to be required at a relatively high concentration for other prebiotically pertinent processes such as protometabolic reactions [17,18], ribozyme activity [19,20,21,22], and nonenzymatic template-directed replication of RNA [23,24]. Therefore, it is necessary to systematically evaluate the metal ion stability of plausible prebiotic compartments to characterize their compatibility with the functioning of other prebiotic processes.

The influence of the overall ionic strength of an aqueous medium on prebiotic membrane-assembly and stability also has implications for discerning the plausible niche(s) where life might have originated on the early Earth [15]. Oceanic hydrothermal vents and terrestrial hot springs are the two widely considered niches for the origins of life on Earth [25,26]. However, the high salinity of seawater would have been detrimental for the prebiotic compartmentalization process [27,28]. In contrast, terrestrial fresh water bodies, such as hot springs and lakes, with their overall low ionic strengths, would have provided a more conducive environment for the membrane-assembly of protoamphiphiles [6,29].

Several decades of laboratory experiments with fatty acid vesicles using low ionic strength buffers provide a proof of concept to support this notion. Importantly, some of the recent advances in the field, wherein fatty acid-based systems have been shown to form vesicles in hot spring water samples collected from early Earth analogue sites, have also strongly supported the above mentioned hypothesis [27,30]. Moreover, these analogue experiments demonstrate the ability of these protoamphiphiles to assemble into vesicles in a natural environment, such as a hot spring, which contains a diverse set of ions in varying concentrations. These natural settings would likely be a closer mimic of a complex prebiotic soup than the stringently controlled laboratory buffered conditions [31]. Therefore, the membrane-assembly of protoamphiphiles under such natural analogue conditions further underscores their relevance and plausibility of constituting protocellular systems on the prebiotic Earth.

Nonetheless, fatty acids might not have been the only option that early protocellular systems would have explored to form compartments. Both meteoritic delivery and endogenous synthesis reactions, along with other amphiphile-co-solute interactions in a prebiotic soup, would have generated a diverse set of protoamphiphiles on the early Earth. Importantly, vesicles made of a heterogeneous pool of amphiphiles have been shown to be highly stable in the presence of metal ions. For example, mixed membranes composed of decanoic acid with other amphiphiles likes decanol, glycerol 1-monodecanoate, and decylamine are stable both in the presence of metal ions as well as under sea salt conditions [28,32,33,34]. Blended vesicles of fatty acids and phospholipids are also shown to be more robust than pure fatty acid vesicles in the presence of magnesium, hinting at the intermediate step towards protocell membrane evolution [35,36]. In addition to fatty acids, other admixtures of SCAs like cyclophospholipids and fatty alcohols also result in robust vesicles that can tolerate high concentrations of divalent cations [37].

Recently, we reported the formation of N-acyl amino acids in a lipid-amino acid mixture under prebiotically relevant wet-dry cycling conditions, and also showed that some of these NAAs readily self-assemble into vesicles at acidic pH [12]. Their vesicle formation behavior was also found to be temperature-dependent. Therefore, a high temperature of 60 °C should to be maintained to keep these amphiphiles in a vesicle form (see Section 2.2.1. for further details). NAA is an interesting class of protoamphiphiles containing an amino acid covalently linked to a fatty acid via an amide linkage (Appendix A). However, despite the undeniable potential of this unique amphiphile to serve as a protocell compartment, the self-assembly behavior and different physicochemical properties of NAA-based membranes are still largely unexplored. Furthermore, the effect of metal ions on these membranes has not been studied yet. Moreover, it is not known whether NAA-based amphiphile systems can self-assemble into membranes under natural aqueous conditions, such as in hot springs, which are characterized by a diversity of ions and ionic strengths.

Here, we systematically evaluated the effect of both divalent and monovalent metal ions on the stability of NAA-based vesicular systems. For this, we used N-oleoyl glycine-based model vesicles, and Mg^2+^ and Na^+^ as the representative divalent and monovalent metal ions, respectively. We show that pure NOG vesicles can sustain Mg^2+^ concentrations in the order of up to 1:1 molar ratio, beyond which large magnesium-induced aggregates are formed. Interestingly, the addition of surfactants like Glycerol 1-monooleate to the NOG system results in a heterogeneous population of vesicles and droplets, which tolerates even higher concentrations of Mg^2+^. These NOG-based systems are even more resistant to Na^+^ ions and typically form aggregates only at a 100-fold higher concentration of Na^+^ as compared to that of Mg^2+^. Both pure and mixed NOG systems are able to assemble into vesicles in hot spring water samples containing multiple ions in varying concentrations. This highlights their potential to form membrane compartments under natural, fresh water hydrothermal conditions, underscoring the relevance of these terrestrial niches in facilitating prebiotic reactions pertinent to the emergence of early cellular life.

## 2. Materials and Methods

### 2.1. Materials

All the amphiphiles (N-oleoyl glycine, Glycerol 1-monooleate, and oleyl alcohol (OOH)) were purchased from Avanti polar lipids (Alabaster, AL, USA). EDTA disodium salt dihydrate and tri-sodium citrate (Qualigens) were procured from HiMedia (Mumbai, India) and Thermo Fisher Scientific (Mumbai, India), respectively. The rest of the chemicals and reagents, including acetic acid and sodium acetate (components of acetate buffer), as well as salts like magnesium chloride (MgCl_2_) and sodium chloride (NaCl), were from Sigma-Aldrich (Bangalore, India). All the reagents were of analytical grade and used without further purification.

### 2.2. Methods

#### 2.2.1. Vesicle Preparation

For the vesicle formation by NOG-based amphiphilic systems, a previously standardized protocol was used [12]. Typically, in metal ion stability reactions containing the pure NOG system, a master mix solution was prepared by taking an appropriate volume of NOG methanol stock (10 mg/mL) in a microcentrifuge tube. For NOG-surfactant mixed systems, appropriate volumes of 10 mg/mL methanol stocks of NOG and GMO (or OOH in some reactions) were mixed in a microcentrifuge tube. The methanol was evaporated under vacuum to form a dry lipid film, which was then hydrated with an appropriate volume of 200 mM acetate buffer pH 5 (prepared in ultrapure water (18.2 MΩ-cm)) to get a final total amphiphile concentration of 7.5 mM in the master mix reaction. This contained either NOG (pure system) or NOG + surfactant in 2:1 ratio (mixed system), unless and otherwise mentioned. This solution was then incubated at 60 °C for 1 h with constant shaking at 500 rpm, with intermediate mixing by vortexing and pipetting. Note that the heating step is very crucial for the vesicle formation by NOG-based amphiphilic systems, as the chain-melting transition temperature of NOG seems to be higher than room temperature [12]. Moreover, this high temperature should be maintained throughout the course of the experiment, in order to keep these systems in a vesicular form and to avoid NOG crystal formation [12]. This was achieved by incubating NOG-based amphiphile solutions in a thermomixer (ThermoMixer C; Eppendorf India) equipped with a temperature maintaining lid (ThermoTop; Eppendorf India) that also avoids volume changes due to evaporation-condensation during the incubation period.

#### 2.2.2. Setting Up the Metal Ion Stability Experiments

Briefly, 40 µL aliquots of the NOG master mix vesicular solution were taken into separate microcentrifuge tubes. To these individual tubes, appropriate volumes of metal ion stock solution were externally added to get increasing concentrations of metal ion (usually in the range of 0 to 11 mM for Mg^2+^ and 0 to 600 mM for Na^+^), while keeping the NOG concentration constant. After adding the metal ion stock solution into individual reaction vials, the final volume was adjusted to 50 µL using 200 mM acetate buffer pH 5, to get a 6 mM working concentration of NOG. Control reaction contained only 6 mM NOG with no externally added metal ions. A similar procedure was followed for the experiments with the NOG + GMO mixed system. For the Mg^2+^ stability experiments, 100 mM MgCl_2_ stock solution was used, while for the Na^+^ stability experiments, 4 M NaCl stock solution was used. Both these stock solutions were prepared in 200 mM acetate buffer pH 5. After the addition of metal ions to vesicular solutions, the reactions were incubated for 30 min at 60 °C to allow the metal ion to interact with vesicles and these samples were then subjected to microscopy.

#### 2.2.3. Microscopic Analysis

All the amphiphile-based higher order structures like vesicles, droplets, and metal ion-induced aggregates that could result in a solution, were visualized using differential interference contrast (DIC) microscopy (Axio Imager Z1, Carl Zeiss, Germany) under 40× objective (NA = 0.75). For the experiments characterizing vesicle formation by NOG-based amphiphile systems in hot spring water samples, the vesicular structures were further confirmed by staining vesicles with an amphiphilic dye and visualizing using epifluorescence microscopy. For this, vesicles were stained with 10 µM of Octadecyl Rhodamine B Chloride (R18) (Invitrogen, Thermo Fisher Scientific; India), which was generously donated by Pucadyil lab at IISER Pune. The staining was performed by adding an appropriate volume of R18 dye methanol stock (50 µM) while making the dry lipid film of the NOG-based amphiphiles. The fluorescent vesicles were visualized using filter set 43 HE (Ex: 550/25 nm, Em: 605/70 nm, Beamsplitter: FT 570), with rest of the settings being same as mentioned above for the DIC microscopy. Image acquisition was done using AxioVision software. Imaging of all the heated reaction solutions was performed as quickly as possible to avoid NOG crystal formation.

#### 2.2.4. Vesicle Re-Formation by the Addition of Magnesium Chelators

Briefly, 6 mM pure NOG and NOG + GMO (2:1 ratio) vesicular solutions containing externally added 12 mM Mg^2+^ ions were prepared as mentioned above. This Mg^2+^ concentration was selected to ensure the formation of magnesium-induced aggregates in both the pure NOG and the NOG + GMO mixed systems. Firstly, the presence of aggregates was confirmed by DIC microscopy. Then, a chelator (EDTA or citrate) was added in 1:1 mole equivalents to that of Mg^2+^ to the same aggregate containing solution, followed by the incubation at 60 °C for 30 min. The dissociation of aggregates and the re-appearance of vesicles was monitored using DIC microscopy.

#### 2.2.5. Vesicle Formation by NOG-Based Amphiphile Systems in Hot Spring Water Samples


I.The collection of water samples from early Earth analogue hot spring sites: Water samples were directly collected from two hot spring pools at Hells Gate Geothermal Reserve, Tikitere (abbreviated as TIKB and TIKC, respectively). These samples were filtered with a 0.22 µm polyethersulfone membrane filter (rinsed with 20 mL of sample) and stored in acid-washed high-density polyethylene bottles until further use. Samples were untreated and unpreserved in the field.II.Geochemical analysis of hot spring water samples: The collected water samples were analysed for the identification and quantification of different ions present in them. These measurements were performed by following standard protocols as reported earlier [38]. Briefly, the concentrations of the major cations (Na^+^, K^+^, Ca^2+^, Mg^2+^, and NH^4+^) and anions (Cl^−^, F^−^,SO_4_^2−^, and NO₃^−^) were measured using ion chromatography instrument Compact IC plus 882 (Metrohm, Herisau, Switzerland). Moreover, the alkalinity of the samples was measured using an auto-titrator Eco Titrator (Metrohm, Switzerland). The dissolved Silica concentrations were measured by conventional molybdenum-blue method using double beam UV-VIS Spectrophotometer (M.D.T. INTERNATIONAL, Ambala, Haryana, India). The accuracy and precision of these analyses were monitored regularly, which had an average value of ±4%. The net inorganic charge balance (NICB) for these samples was around 1, which indicates good data quality.III.Vesicle formation in hot spring water samples: Dry lipid films of pure NOG and mixed systems containing NOG + surfactant (GMO or OOH) in 2:1 ratio were prepared as previously described. These lipid films were hydrated with 50 µL of hot spring water sample (TIKB/TIKC) to get a total amphiphile concentration of 6 mM. The solution was further incubated at 60 °C for 1 h with constant shaking at 500 rpm, with intermittent mixing by vortexing and pipetting to facilitate the vesicle formation process. The initial pH of the hot spring water samples was 7–7.5, which decreased to 5–5.5 after the addition of the amphiphiles, likely because of the acidic nature of NOG. The formation of vesicles was checked by both DIC and epifluorescence microscopy as mentioned above.


All the experiments in this entire study were performed in at least three independent replicates.

## 3. Results

### 3.1. Effect of Mg^2+^ on the Stability of Pure NOG and NOG + GMO Mixed Systems

In our previous study, we showed that 6 mM NOG in 200 mM acetate buffer pH 5 results in the formation of a large number of vesicles [12]. Therefore, these conditions were used to evaluate the effect of different metal ions on the stability of NOG vesicles. We started this characterization with divalent metal ions using magnesium (Mg^2+^) as a representative ion, because of its direct relevance to several prebiotic processes pertinent to the RNA world. As mentioned earlier, it is not only crucial for RNA folding and ribozyme activity [19,20], but also required for the nonenzymatic template-directed replication of RNA [39]. To test the effect of Mg^2+^ on NOG vesicles, NOG concentration was kept constant at 6 mM and the Mg^2+^ concentration was varied incrementally up to 11 mM. For the pure NOG system, it was observed that NOG vesicles could tolerate Mg^2+^ concentrations up to 1:1 molar ratio, beyond which magnesium ions induced a clumping of vesicles resulting in the formation of large-sized aggregates (Figure 1a and Appendix A).

The morphology of these metal ion-induced aggregates was different from the NOG crystals that form at a lower temperature (Appendix A). Overall, the behavior of the pure NOG system in the presence of increasing concentrations of Mg^2+^ followed three distinct phases (Figure 1c). A vesicular phase was observed up to 5 mM Mg^2+^, which contained only vesicles with no visible aggregates. It was followed by a transition phase at 6 mM Mg^2+^, where small aggregates started forming in the solution, and both free vesicles and aggregates were present simultaneously. Finally, an aggregate phase was observed from 7 mM Mg^2+^ onwards, where the vesicular system formed large aggregates, with no free vesicles present in the solution.

Surfactants like monoglyceride and fatty alcohol are known to increase the stability of fatty acid vesicles towards metal ions [28,34]. Therefore, we sought to explore whether a similar effect is observed for NOG-surfactant mixed systems also. Firstly, the optimum NOG-surfactant mixed system to perform this experiment was narrowed down by systematically evaluating the vesicle formation behavior of different combinations of NOG-surfactant binary systems by varying their ratios. Surfactants used for these experiments were GMO and OOH with different NOG to surfactant ratios, including 1:1, 2:1, and 4:1, while the total amphiphile concentration was kept constant at 6 mM. In NOG + GMO mixed systems, it was observed that, irrespective of the ratios, a mixture of NOG and GMO always generated a heterogeneous population of vesicles and droplets (Appendix A; appendix A of our earlier study [30] details the criteria that we used to distinguish between vesicles and droplets using DIC microscopy). Similar results were observed even at a lower total amphiphile concentration of 3 mM (Appendix A). Nonetheless, a mixed system of NOG + GMO with 2:1 ratio was found to be optimum for studying the metal ion effect. This allowed the formation of large-sized vesicles as compared to the 1:1 ratio system, making it easy to observe them under the microscope (Appendix A). Furthermore, this 2:1 ratio also conferred greater stability in the presence of higher Mg^2+^ concentrations in comparison to the 4:1 ratio (Appendix A). In the case of the NOG + OOH system, we observed that OOH had a destabilizing effect on the NOG system, which predominantly led to droplet formation at all the three NOG to OOH ratios (Appendix A). Therefore, this system was not evaluated further for its metal ion stability.

With NOG + GMO (6 mM; 2:1 ratio) mixed system, it was observed that the addition of GMO indeed increased the stability of NOG towards magnesium ions as this mixed system was able to tolerate Mg^2+^ concentrations till 11 mM (Figure 1b). Consistent with the pure NOG system, this mixed system also showed a three-phase behavior in the presence of increasing Mg^2+^ concentrations (Figure 1c). A vesicular phase containing a mixture of vesicles and droplets was observed up to 7 mM Mg^2+^. It was followed by a transition phase from 8 to 10 mM Mg^2+^, where small metal ion-induced aggregates were also present along with vesicles and droplets. Finally, the aggregate phase appeared from 11 mM Mg^2+^ onwards, where the system completely flocculated into large-sized aggregates. It is pertinent to note that the span of both the vesicular and transition phases of the NOG system increased after the addition of GMO to the system, which clearly indicated a stabilizing effect that was being conferred by the surfactant towards the metal ions. We also note that some variability in the aggregation behavior is expected to be observed in the transition phase, especially for the NOG + GMO mixed system, as it is difficult to achieve the exact same ratio of NOG to GMO in every resultant vesicle. This would essentially render vesicular systems with a lower concentration of GMO more vulnerable to metal ion-induced aggregation (Appendix A).

After evaluating the effect of Mg^2+^ on the stability of pure NOG and the NOG + GMO mixed systems, we tested whether the aggregates observed in the presence of Mg^2+^ were indeed magnesium-induced aggregates, and whether this aggregation phenomenon is reversible. To discern this, we added EDTA to the solution in 1:1 mole ratio to Mg^2+^, after the formation of the aggregates. EDTA chelated Mg^2+^ ions in the solution, which resulted in the disruption of the large-sized aggregates and reappearance of vesicles in both the pure NOG and NOG + GMO mixed systems (Figure 2a). EDTA is an artificially synthesized chelating agent that is used majorly in commercial applications. However, there are many naturally occurring chelating agents that can also effectively chelate magnesium ions.

Given the scope of this study, we asked whether there are any prebiotically plausible chelating agents that might have provided further stability to protocell membranes in the presence of metal ions. Citrate is considered as an important metabolite both in extant as well as in proto-metabolic pathways [40], and is known for its chelation effect [41,42]. Therefore, we tested the effect of citrate on magnesium-induced aggregates and observed similar results as those with EDTA, where vesicles emerged from large aggregates (Figure 2b).

### 3.2. Effect of Na^+^ on the Stability of Pure NOG and NOG + GMO Mixed Systems

After testing the effect of a divalent metal ion on the stability NOG-based systems, we aimed to characterize the behavior of these systems in the presence of a monovalent cation. For this, we used sodium (Na^+^) as the representative ion. Previous studies have shown that fatty acid-based systems are more stable in the presence of monovalent cations than divalent ones [28]. We observed a similar behavior for NOG-based systems as well, where pure NOG vesicles formed aggregates only at and above 400 mM Na^+^ (Figure 3), which is about two orders of magnitude higher than that for Mg^2+^ (7 mM). The NOG + GMO mixed system was even more stable and resulted in aggregate formation only at 500 mM Na^+^ (Figure 3).

In addition to aggregate formation, another intriguing phenomenon was observed in the case of the pure NOG system, wherein the addition of 100 mM Na^+^ resulted in vesicle shrinkage and also the formation of small-sized droplets in solution (Figure 3 and Appendix A). This effect was consistent even for higher concentrations of Na^+^ (i.e., at 200 mM and 300 mM). We hypothesized two plausible reasons for this effect. (1) The vesicle shrinkage might have occurred because of an osmotic shock faced by vesicles upon the external addition of NaCl to the solution. NOG vesicles were prepared in an acetate buffer that already contained around 143 mM Na^+^ that was added during the buffer preparation process. Given that these vesicles are not readily permeable to charged species, the external addition of 100 mM Na^+^ to a solution containing preformed vesicles would have generated a higher concentration of Na^+^ in the exterior as compared to that of the vesicle lumen. This would have resulted in the outflow of water from vesicles causing the overall shrinkage of these vesicles, a phenomenon similar to plasmolysis. To validate this hypothesis, we performed an experiment where the Na^+^ concentration in the acetate buffer was increased by 100 mM and this buffer with higher [Na^+^] was used for vesicle preparation. The vesicle shrinkage in this reaction was not as dramatic as what was observed in the reaction with externally added Na^+^ (Appendix A). This was likely because the Na^+^ levels in the vesicle exterior and in the lumen would have gotten equilibrated during the vesicle formation process itself. Pertinently, it has been previously observed in the case of fatty acid systems that the vesicles undergo an osmotic shock upon the external addition of salts to a solution containing preformed vesicle [28]. Notably, this overall vesicle shrinkage phenomenon was less apparent in the NOG + GMO mixed system (Figure 3). It is known that the mixed membranes made of fatty acids and monoglycerides have a higher permeability than the pure fatty acid vesicles [7,34]. It is very possible that the presence of GMO increases the permeability of NOG + GMO mixed vesicles when compared to pure NOG vesicles. This probably would have allowed the faster equilibration of Na^+^ between vesicle lumen and the exterior, thereby reducing the exosmosis of water and the resultant vesicle shrinkage. (2) The formation of droplets in the pure NOG system after the addition of Na^+^ could be because of the charge screening of the deprotonated terminal carboxyl moiety (COO^−^) of the NOG head group by Na^+^ ions. This would make the overall system more hydrophobic, which could have resulted in the droplet formation. In such a case, a similar effect should also be observed with other monovalent cations. In order to test this hypothesis, we externally added 100 mM K^+^ ions to the solution containing preformed NOG vesicles, which also resulted in droplet formation along with vesicle shrinkage similar to what was observed in the Na^+^ reaction (Appendix A).

### 3.3. NOG-Based Amphiphile Systems Readily Form Vesicles in Hot Spring Water Samples

After systematically evaluating the stability of NOG-based systems in the presence of both monovalent and divalent metal ions, we set out to explore the self-assembly behavior of these systems in the presence of multiple cations and anions of different concentrations. This was to get a realistic sense of how NAA-based amphiphilic systems would behave in natural water bodies that often possess multiple ions in varying concentrations. To test this, we used water samples that were collected from hot spring sites, and we did this for two reasons. Firstly, mimicking such a system with diverse ionic strength in the laboratory is difficult, as these ions are added in the form of salts, which invariably results in the excess addition of some counter ions while preparing such ionic mimics. Secondly, and pertinently, terrestrial hot springs are considered as one of the plausible niches where life would have potentially originated on early Earth and Mars [26]. Therefore, studying prebiotic membrane-assembly processes by using water samples from contemporary terrestrial hot spring sites, which have topological features analogous to early Earth and Martian conditions, allows one to glean the feasibility of prebiotic compartmentalization under more realistic, diverse, and extreme environments. We have previously shown that fatty acid-based systems can assemble into vesicles in hot spring water [30], which motivated us to check whether a similar phenomenon could be observed in case of NAA-based systems too.

To perform these analogue experiments, we used water samples collected from two hot spring sites in Tikitere, New Zealand, which are abbreviated as TIKB and TIKC, and analysed for their ionic content (Table 1). Both TIKB and TIKC samples had a dominance of NH₄⁺ and SO₄^2−^ ions. Several other cations (Na⁺, K^+^, Ca^2+^, Mg^2+^) and anions (Cl^−^, F^−^, NO₃^−^, HCO₃^−^) were also detected in varying concentrations in these water samples, highlighting the ionic diversity that such natural aqueous systems possess. However, it is important to note that the concentrations of both monovalent and divalent cations were far below the level at which they generally induce vesicle aggregation in laboratory settings.

It was observed that both pure NOG (6 mM) and NOG + GMO (6 mM; 2:1 ratio) mixed systems readily assembled into vesicles in both TIKB and TIKC samples (Figure 4). The resultant vesicle population was heterogeneous in terms of the size, shape and lamellarity of vesicles. Therefore, for better visualization of such higher order structures, these vesicles were stained using octadecyl rhodamine B chloride (R18) (an amphiphilic dye that readily partitions into membranes) and observed under epifluorescence microscopy, which further delineated the lamellarity of these vesicles. We further observed that a mixture of NOG + OOH (6 mM; 2:1 ratio) also resulted in vesicle formation in TIKC (Appendix A). Although the vesicle formation by lipid film hydration method is known to generate a morphologically diverse set of vesicles, the variety of ions and their relative concentrations in these hot spring samples could further affect the self-assembly and the morphology of these vesicles.

## 4. Discussion

N-acyl amino acid is a hybrid molecule comprising both fatty acid and amino acid moieties within the same structure. This amalgamation provides this amphiphile with great flexibility in terms of its being able to acquire different properties based on the varying nature of the amino acid head group, to better adapt to the fluctuating and extreme environmental conditions thought to have been prevalent on the prebiotic Earth. For example, the addition of glycine as a head group to the carboxyl terminal of oleic acid (OA), generates N-oleoyl glycine that can form vesicles at acidic pH, whereas OA itself is unable to do so. Thus, NAAs might have played a pivotal role towards bringing in the necessary diversity and adaptability in protocellular compartments, which is one of the fundamental requisites for the advent of Darwinian evolution. There have been few attempts to functionalize fatty acid membranes by doping them with small amounts of NAAs [44,45]. However, to the best of our knowledge, NAA itself as an amphiphilic system has not been systematically investigated in a prebiotic context, and the effect of ions on their self-assembly and stability is not yet known.

In this study, we evaluated the behavior of NOG-based model vesicular systems in the presence of metal ions, given the central importance of metal ions towards facilitating several pertinent processes in the RNA world [19,20,39]. Although the addition of an amino acid as a head group could potentially alter the basic physicochemical properties of fatty acids as mentioned earlier, especially in terms of the optimum pH for their vesicle formation, some of the features of NAAs and fatty acids may still be comparable. Herein, we demonstrate that the metal ion stability of these amphiphiles is one such feature. Previous studies exploring the effect of Mg^2+^ on fatty acid membranes show that vesicles made of only OA form large metal ion-induced aggregates when the Mg^2+^ concentration exceeds the fatty acid concentration [34,35]. We report a similar phenomenon with vesicles made of NOG (an NAA analogue of oleic acid), wherein these vesicles form large aggregates when the Mg^2+^ concentration just surpasses the NOG concentration. Furthermore, the NOG system was found to be much more stable in the presence of Na^+^ than of Mg^2+^, comparable to what has been reported for fatty acids [28]. Moreover, NOG vesicles undergo an osmotic shock upon the external addition of NaCl to the solution, which is consistent with previous observations with fatty acid vesicles [28].

This significant similarity between NAA and fatty acid vesicles in terms of their stability towards metal ions may likely stem from the nature of their terminal ionizable moiety. The behavior of an amphiphile that has an anionic head group in the presence of metal ions is primarily governed by the electrostatic interaction of its head group with the positively charged metal ion. Given that both fatty acids and NAAs have a carboxyl terminus (-COOH), which can deprotonate to form a carboxylate ion (-COO^−^) depending on the pH of the solution, they show a comparable behavior in terms of their metal ion stability.

The incompatibility of SCA vesicles with high concentrations of metal ions (particularly divalent cations), which are required for nonenzymatic RNA strand copying and for ribozyme catalysis, has been considered a major obstacle towards the emergence of protocells [39]. One approach to overcome this difficulty is to systematically investigate different membrane compositions of prebiotically pertinent amphiphiles to discern which compositions generate stable membranes at high metal ion concentrations. This approach is also logical and realistic since a prebiotic soup would have likely harbored a plethora of amphiphiles, which inevitably would have generated compositionally heterogeneous membranes on the early Earth [9]. Interestingly, prebiotically plausible amphiphiles with a polar non-ionic head group such as monoglycerides have been shown to stabilize fatty acid vesicles in the presence of both divalent and monovalent cations [28,34]. Our study shows that such a stabilizing effect can also be achieved in the case of NAAs by generating blended membranes of NAAs and monoglycerides. Particularly, mixed vesicles composed of NOG and GMO were highly stable in the presence of both Mg^2+^ and Na^+^ as compared to those made of only NOG.

Another possibility to tackle the above mentioned problem is to look for prebiotically plausible chelators that could form a complex with a metal ion in such a way that the coordination complex still remains functionally active to facilitate previously mentioned RNA-related reactions [39]. In such a scenario, it is possible to achieve low levels of free metal ions in the solution, which allow SCAs to sustain their vesicular form. In this context, citrate is an important molecule as it has an excellent chelating capacity and also a key metabolite in both extant as well as proto-metabolic pathways. Adamala et al. showed that citrate can chelate magnesium while still allowing nonenzymatic RNA template copying inside intact fatty acid vesicles at alkaline pH regime (pH 8) [42]. Given the influence of pH on the chelation ability of a chelator, our study demonstrates that citrate can effectively chelate magnesium even in the acidic regime of pH 5. The addition of citrate reversed the magnesium-induced aggregation phenomenon and allowed for the re-formation of vesicles, both in the pure NOG and the NOG + GMO mixed systems. It will be worthwhile to further evaluate whether NAA-based vesicular systems containing magnesium-citrate complex are compatible with RNA formation and ribozyme catalysis reactions.

Thus far, we discussed the adverse effect of metal ions on the stability of SCA vesicles. However, an equally important factor that is often neglected is that metal ions can actually facilitate, and are important for, vesicle self-assembly processes, albeit at a low concentration [46]. They can do so by salting-in the hydrophobic tails and stabilizing the negatively charged carboxylate moieties of the head group via electrostatic interactions. Therefore, from a prebiotic perspective, natural fresh water systems with their overall low ionic strengths would have been more favorable for a protocell compartmentalization process as compared to marine niches [47]. In this study, we showed that NOG-based model vesicular systems readily assemble into vesicles in hot spring water samples. These results have two-fold implications. Firstly, they demonstrate the ability of NAAs to assemble into vesicles in natural aqueous systems with varying ionic strengths, which further substantiates their candidature as model protocell membranes. Secondly, these results provide yet another piece of evidence for the conduciveness of terrestrial freshwater systems for the self-assembly of prebiotic compartments, in addition to supporting nonenzymatic oligomerization processes [31,48]. This, in turn, underscores the likelihood of these niches as plausible hatcheries for the origin and early evolution of life on Earth.

Overall, we discerned the influence of ions on the self-assembly and stability of NAA-based vesicular systems. This is an important parameter that needed to be systematically evaluated as researchers in the “Origins of life” field continue to work with this intriguing amphiphile system as a model protocell membrane and its plausible role in facilitating other prebiotically pertinent processes. It is worthwhile to mention in this context that several important aspects of this unique amphiphile, such as its stability under fluctuating conditions of temperature and pH, or its potential to catalyze different chemical reactions on the membrane surface via the side chain of its amino acid head group, are yet to be systematically explored. Future research in this direction will enable shedding greater light on the adaptability and functionality of early protocellular compartments in a prebioloical era.

## Figures and Tables

**Figure 1 life-11-01413-f001:**
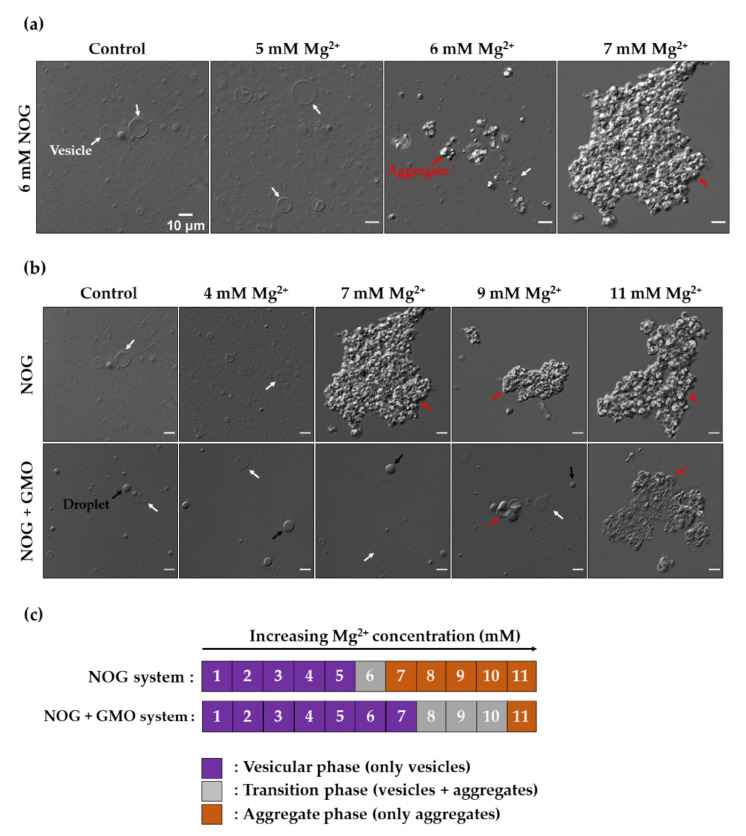
Effect of Mg^2+^ on pure NOG and NOG + GMO mixed vesicles. (**a**) NOG vesicles are stable in the control reaction (no externally added Mg^2+^) and in the presence of 5 mM Mg^2+^. However, metal ion-induced aggregates appear in the reaction containing 6 mM Mg^2+^, along with free vesicles. Finally, in the presence of 7 mM Mg^2+^, vesicles completely collapse into large aggregates and no free vesicles are observed. (**b**) Addition of GMO to NOG system increases the stability of vesicles towards Mg^2+^ ion. 6 mM NOG system completely collapses in the presence of 7 mM Mg^2+^ and above. However, it takes 11 mM Mg^2+^ to induce large aggregates in the case of NOG + GMO (6 mM; 2:1 ratio) mixed system. Notably, NOG + GMO mixture results in a heterogeneous population of vesicles and droplets unlike pure NOG system, which predominantly forms vesicles. Vesicles, droplets and aggregates are indicated by white, black, and red arrows respectively. Imaging was done using DIC microscopy. Scale bar is 10 µm for all microscopy images, unless mentioned otherwise. (**c**) A distinct three-phase behavior of NOG-based vesicles is depicted with increasing concentrations of Mg^2+^. The vesicular phase (purple) contains only vesicles and no metal ion-induced aggregates. It is followed by a transition phase (grey) where both free vesicles and small metal ion-induced aggregates are simultaneously present. Finally, the aggregate phase (orange) contains only large-sized aggregates. Although, the color code of the three phases has been described in terms of vesicles and aggregates, the NOG + GMO mixed system also contains droplets in addition to vesicles.

**Figure 2 life-11-01413-f002:**
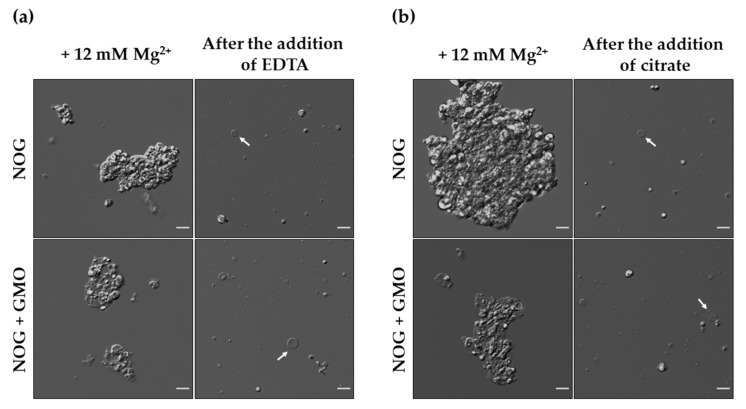
Re-formation of vesicles from magnesium-induced aggregates after the addition of a chelator. (**a**) Both pure NOG (6 mM) and NOG + GMO (6 mM; 2:1 ratio) mixed systems (top and bottom panels respectively) completely collapse into large aggregates in the presence of 12 mM Mg^2+^. However, the addition of EDTA in 1:1 mole equivalents to that of Mg^2+^ results in the disassembly of aggregates with a concurrent reappearance of free vesicles (indicated by white arrows). (**b**) Similar effect is observed in the presence of citrate as a chelator, which was also added in 1:1 mole equivalents to that of Mg^2+^. Imaging was done using DIC microscopy. Scale bar is 10 µm.

**Figure 3 life-11-01413-f003:**
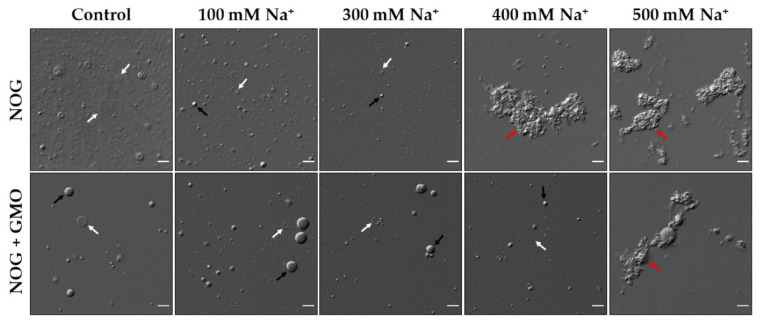
Effect of Na^+^ on the stability of NOG-based vesicles. 6 mM pure NOG vesicles tolerate Na^+^ concentrations up to 300 mM, beyond which large metal ion-induced aggregates are formed (top panel). Also, the external addition of Na^+^ to pure NOG vesicular system results in vesicle shrinkage and the formation of small droplets as observed in the 100 mM and 300 mM Na^+^ reactions. The NOG + GMO (6 mM; 2:1 ratio) mixed system (bottom panel) forms a heterogeneous population of vesicles and droplets, which survives Na^+^ concentrations up to 400 mM, beyond which metal ion-induced aggregates are formed. Vesicles, droplets and aggregates are indicated by white, black and red arrows respectively. Imaging was done using DIC microscopy. Scale bar is 10 µm.

**Figure 4 life-11-01413-f004:**
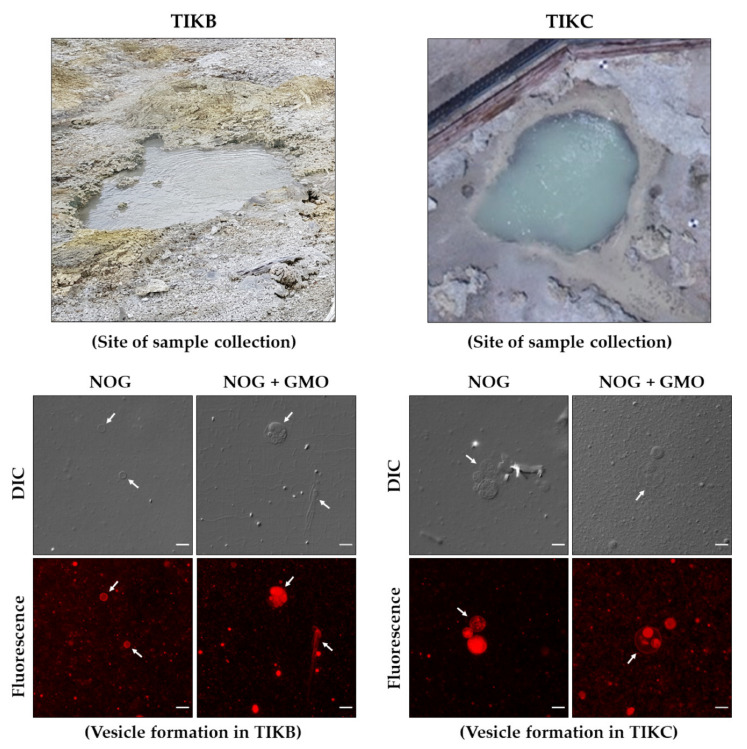
Vesicle formation by NOG-based amphiphile systems in hot spring water samples. Both pure NOG (6 mM) and NOG + GMO (6 mM; 2:1 ratio) mixed systems readily form vesicles (indicated by white arrows) in two different hot spring water samples with acronyms TIKB (**left panel**) and TIKC (**right panel**). Actual sites of sample collection are shown in top panels, while microscopy images for the vesicle formation in respective hot spring samples are shown in bottom panels. Vesicle morphology varies from unilamellar to multivesicular vesicles, which were visualized under DIC as well as fluorescence microscopy. For fluorescence imaging, vesicles were stained with an amphiphilic dye named octadecyl rhodamine-B chloride (R18). Fluorescence images are pseudocolored for a better visualization. Scale bar is 10 µm. TIKC sample collection site image was adapted from [43].

**Table 1 life-11-01413-t001:** Geochemical analysis of the hot spring water samples that were used in this study. TZ⁺ and TZ^−^ indicate sum of cations and sum of anions respectively in microequivalent units. TZ⁺/TZ^−^ is net inorganic charge balance (NICB).

Hot Spring Sample	Major Cations					TZ⁺	Major Anions					TZ^−^		TZ⁺/TZ^−^
	NH₄⁺	Na⁺	K⁺	Ca^2^⁺	Mg^2^⁺		NO₃^−^	F^−^	Cl^−^	HCO₃^−^	SO₄^2−^		SiO₂	
	All Values in µM					µE	All Values in µM					µE	µM	
TIKB ^†^	2756	743	285	98	31	4043	24	45	122	29	2550	5319	1128	0.76
TIKC ^†^	3296	1007	301	167	27	4991	14	48	150	1458	2312	6294	1019	0.79

^†^ The initial pH of TIKB and TIKC was 7–7.5, which decreased to 5–5.5 after the addition of the amphiphiles (likely because of the acidic nature of NOG).

## Data Availability

Not Applicable.

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
