# Peer review of "Influence of Metal Ions on Model Protoamphiphilic Vesicular Systems: Insights from Laboratory and Analogue Studies"

_life, 2021, doi:10.3390/life11121413_

Round 1

Reviewer 1 Report

This manuscript describes the effect of metal ions on oleoyl glycine-based liposomes. The findings are definitely novel. How relevant this specific system is to explain the origins of life is less clear. For example, why should one focus on oleoyl derivatives, when it is widely believed that prebiotic lipids were most likely saturated or randomly/polyunsaturated (the insertion of a double bond in position C9-C10 has never been shown in a prebiotic context)? Or why should one try to find alternative amphiphiles to form liposomes in slightly acidic pH values (there are already examples of amphiphiles capable of self-assembling in acidic pH values)? However, the experiments are nicely performed and the use of acylated amino acids is intriguing.

The title should include "metal" before "ions". The authors are mainly looking at the effect of Mg and Na, rather than other organic ions.

The introduction is missing many relevant publications that should be at least cited, if not discussed.

Line 36: work from the Sutherland group on phosphatidic acid synthesis should be included (JACS 2019).

Line 38: a reference is needed.

Lines 44-45: the authors mention temperature as one of the variables that affect the stability of liposomes. The authors should cite Mansy and Szostak (PNAS 2008) and Rubio-Sanchez et al (JACS 2021).

Lines 49-50: the authors should include work from the Mansy group (e.g., Nat. Catal. 2018) for metal-based protometabolism, from the Holliger and the Joyce groups for ribozyme activity and from the Sutherland group for Mn-mediated ligation (Nat Chem 2013).

In general, many groups have explored the effect of metal ions on different types of liposomes (e.g., Sahai's group, Maurer's group, Lane's group, Szostak's group). The authors should include a paragraph at least reporting similar work to that reported in this manuscript on different amphiphiles.

Curiosity: have the authors tried to make decanoyl glycine? My guess is that a shorter chain would allow the authors to work at RT. If working at 60°C is particularly important, this should be clearly stated and explained in the introduction.

Line 139: change "was" with "were".

Line 152: change "mental" with "metal".

Line 396: The authors mention two possible reasons for the noticed effect, but I got lost in the first reason (osmotic pressure) and missed the second. It would be useful to rewrite this paragraph and clearly state (with numbers) what the two hypotheses are.

Line 398: The authors mention that 143 mM Na is already present in solution due to the buffer. This is quite a high concentration already. Since the authors are using 6 mM amphiphile, have they tried decreasing the concentration of the buffer, so that the initial concentration of Na is not already incredibly high?

The liposomes formed in the two different hot spring samples look slightly different in terms of morphology. The author should include at least a brief comment on whether they think it is due to the different ion variety/concentration or to other factors.

Author Response

The response to Reviewer 1's comments has been attached as a doc file.

Reviewer 2 Report

Dear Editor, dear authors,

please find my comments in the attached file "review_ions_vesicles_life.txt"

Author Response

The response to Reviewer 2's comments has been attached as a doc file.
